# Recombinant *Ehrlichia canis* GP19 Protein as a Promising Vaccine Prototype Providing a Protective Immune Response in a Mouse Model

**DOI:** 10.3390/vetsci9080386

**Published:** 2022-07-27

**Authors:** Boondarika Nambooppha, Amarin Rittipornlertrak, Anucha Muenthaisong, Pongpisid Koonyosying, Paweena Chomjit, Kanokwan Sangkakam, Sahatchai Tangtrongsup, Saruda Tiwananthagorn, Nattawooti Sthitmatee

**Affiliations:** 1Laboratory of Veterinary Vaccine and Biological Products, Faculty of Veterinary Medicine, Chiang Mai University, Chiang Mai 50100, Thailand; boondarika.n@cmu.ac.th (B.N.); amarin.r@cmu.ac.th (A.R.); anucharham@gmail.com (A.M.); pongpisid_koo@cmu.ac.th (P.K.); paweena_chomjit@cmu.ac.th (P.C.); kanokwansangkakam@gmail.com (K.S.); 2Department of Veterinary Biosciences and Public Health, Faculty of Veterinary Medicine, Chiang Mai University, Chiang Mai 50100, Thailand; saruda.t@cmu.ac.th; 3Ruminant Clinic, Department of Food Animal Clinics, Faculty of Veterinary Medicine, Chiang Mai University, Chiang Mai 50100, Thailand; 4Department of Companion Animal and Wildlife Clinic, Faculty of Veterinary Medicine, Chiang Mai University, Chiang Mai 50100, Thailand; sahatchai.t@cmu.ac.th; 5Excellence Center in Veterinary Bioscience, Office of Research Administration, Chiang Mai University, Chiang Mai 50100, Thailand

**Keywords:** CME, *Ehrlichia canis*, GP19, mice, recombinant protein, vaccine prototype

## Abstract

**Simple summary:**

One of the limitations of vaccine development against *E. canis* infection is the indefinite knowledge of the protective immunity in the host. In this study, recombinant protein GP19 was produced as a vaccine prototype, rGP19, for inducing protective immune responses in a mouse model against *E. canis*. Antibody responses against *E. canis* were evaluated and revealed that the immunized mice with rGP19 showed higher antibody levels than in adjuvant-immunized and naive mice, both pre- and post-challenging with *E. canis*. DNA from blood, liver, and spleen were extracted to determine ehrlichial loads. The rGP19-immunized mice showed significantly lower ehrlichial loads in blood, liver, and spleen DNA compared with adjuvant-immunized mice. This study also detected IFN-γ-producing CD4+ T cells in the rGP19-immunized mice and then were later infected with *E. canis* on day 14 of the post-infection period using flow cytometry. Additionally, Cytokine mRNA expression was investigated and revealed up-regulation of *IFNG* and *IL1* mRNA expression in rGP19-immunized mice. The present study provides evidence of rGP19 that can eliminate *E. canis* by manipulating both humoral and cell-mediated immune responses in the laboratory animal model.

**Abstract:**

The intracellular bacterium *Ehrlichia canis* is the causative pathogen of canine monocytic ehrlichiosis (CME) in dogs. Despite its veterinary and medical importance, there is currently no available vaccine against this pathogen. In this study, the recombinant GP19 (rGP19) was produced and used as a recombinant vaccine prototype in a mouse model against experimental *E. canis* infection. The efficacy of the rGP19 vaccine prototype in the part of stimulating B and T cell responses and conferring protection in mice later challenged with *E. canis* pathogen were evaluated. The rGP19-specific antibody response was evaluated by ELISA after *E. canis* challenge exposure (on days 0, 7, and 14 post-challenge), and demonstrated significantly higher mean antibody levels in rGP19-immunized mice compared with adjuvant-immunized and naive mice. Significantly lower ehrlichial loads in blood, liver, and spleen DNA samples were detected in the immunized mice with rGP19 by qPCR. The up-regulation of *IFNG* and *IL1* mRNA expression were observed in mice immunized with rGP19. In addition, this study detected IFN-γ-producing memory CD4+ T cells in the rGP19-immunized mice and later infected with *E. canis* on day 14 post-infection period using flow cytometry. The present study provided a piece of evidence that rGP19 may eliminate *E. canis* by manipulating Th1 and B cell roles and demonstrated a promising strategy in vaccine development against *E. canis* infection in the definitive host for further study.

## 1. Introduction

Canine monocytic ehrlichiosis (CME) is one of the major infectious diseases for dogs, globally. CME is a rickettsial disease which leads to a multi-systemic disease in dogs, caused by *Ehrlichia canis*. In addition, there were shreds of evidence of *E. canis* infection reported in wild mice and humans [1,2]. Despite its potential veterinary and medical importance, there is currently no commercial vaccine against *E. canis*. Previous studies have shown partial clinical protection of inactivated and attenuated vaccine candidates after being challenged with the virulent strain [3,4]. One of the disadvantages of primitive inactivated and live-attenuated vaccines is the undesirable effects associated with some of the antigenic proteins, this has led to the need for the development of modern vaccines [5,6]. In *Ehrlichia* spp., recombinant P29 protein was developed and substantiated an ability to protect against *Ehrlichia muris* in mice [6]; however, there is no available information on this type of vaccine against *E. canis* infection.

*Ehrlichia* spp. have been characterized by their reactivity with antibodies from infected hosts, several antigens have been identified [6,7]. One of the noticeably major immunoreactive proteins of *E. canis*, 19-kDa glycoprotein (GP19) is detected with dog sera during the acute phase of canine monocytic ehrlichiosis. The GP19 was highly conserved among the geographically distributed *E. canis* strains [7,8,9,10]. GP19 of *E. canis* was found to have orthologs with variable-length PCR target proteins (VLPT) of *Ehrlichia chaffeensis*, but without the tandem repeats (TRs) that were present in *E. chaffeensis* VLPT [11]. Furthermore, *E. canis* GP19 protein consists of a serine/threonine/glutamate (STE)-rich patch at the amino-terminal that contains a major species-specific antibody epitope recognized by the host [11]. Although this protein seems to play an essential role in host-microbe interaction, the information on an application using GP19 as a vaccine candidate against *E. canis* infection is not available. 

In the present study, recombinant protein GP19 (rGP19) was produced and purified to use as a vaccine prototype against *E. canis* infection in mice. The efficacy of the rGP19 vaccine candidate was investigated in the protection and immune response after mice were challenged with *E. canis* pathogen at the in vivo level. There was no report that *E. canis* causes disease in mice; however, *E. canis* infection in wild mouse and cultivation in BALB/c mouse macrophage cells were reported [1,12]. A mouse model was used in this study for immunization and challenge infection at the laboratory level, indicating the possibility of this vaccine being used in animals, especially dogs which are the definitive host, or for further industrial production. The ehrlichial loads were determined after *E. canis* challenge exposure using real-time PCR. Likewise, the cytokine gene expression in mice derived from GP19 peptide-based vaccine was explored. In addition, the frequencies of *E. canis* rGP19-specific IFN-γ-producing CD4+ and CD8+ of mice immunized with or without rGP19 were investigated as well using flow cytometry.

## 2. Materials and Methods

### 2.1. E. canis Organism Cultivation and Stocks

*E. canis* BF W053712 X + 5 (PTA-5811™, ATCC^®^, Manassas, VA, USA) was cultured in DH82 canine macrophage cells at 37 °C and 5% CO_2_ as previously described [13]. DH82 cells with intracellular bacteria, *E. canis*, were propagated until 80% of infected cells were reached. A number of the 1 × 10^6^ infected DH82 cells were collected and stored at −80 °C until they were used. For infection, ehrlichial stocks were used for experimental inoculation of the mice by the intraperitoneal route (i.p.) as described previously [6].

### 2.2. Mice

The mice used in this study were 6- to 8-week-old female BALB/c 40 mice, which were obtained from the National Laboratory Animal Center at Mahidol University, Thailand. All mice were acclimated for 1 week before starting the experiment. The experiment was carried out under the animal biosafety condition at 21 ± 1 °C with 50 ± 10% relative humidity and a 12 h light and dark cycle. The mice were provided laboratory grade of food and water, ad libitum. The experimental procedures for the mouse model were approved by the Institutional Biosafety Committee (IBC), Faculty of Veterinary Medicine, Chiang Mai University, Chiang Mai, Thailand (IBC Approved no. A-0763001). All experiments using animals were performed at Laboratory Animal Center (Office of Research Administration, Chiang Mai University) with the approval of Animal Use Protocol (AUP) for Permission of Animal Care and Use (code: 2563/MC-0008).

### 2.3. Expression of Recombinant E. canis GP19 Proteins

*Ehrlichia* open reading frame (ORF) *gp19-cm03* (GenBank accession no. MF771088) was directionally cloned into pET30a (Genscript, Piscataway, NJ, USA). The Sequence analysis was performed to verify the appropriate frame of the insert. The plasmids were transformed into BL21 Star™ (DE3) (Invitrogen™, Thermo Fisher Scientific, Waltham, MA, USA) for producing the recombinant protein (rGP19). Briefly, the competent cells with a recombinant GP19 pET30a plasmid, were cultured in Luria–Bertani (LB) broth containing the kanamycin at 37 °C for 12–16 h and shaken at 200 rpm until the cell count reached the optical density (OD) at a wavelength of 600 nm reached 0.5–0.6. Then, Isopropyl-β-D-thiogalactopyranoside (IPTG) solution was added to the cells to a final concentration 0.5 mM and continually shaken at 125 rpm at 16 °C for another 16 h. Cell pellets were then collected by centrifugation at 4000× *g* for 20 min. Only the sedimented cells were collected for protein extraction by lysis solution at a ratio of 5 mL per 1 g of coagulant cells, shaken at 75 rpm at room temperature for 60 min, and centrifuged at 10,000× *g* for 20 min at 4 °C. The solution was collected for protein extraction.

For the vaccine prototype, rGP19 was purified by Ni-NTA affinity chromatography using HisTrap HP columns (GE Healthcare Life Sciences, Piscataway, NJ, USA). The protein size was then isolated by electrophoresis (NATIVEN®; ATTO Gentaur, Tokyo, Japan). SDS-PAGE and Western immunoblotting were performed to verify the rGP19 molecular mass and the HRP binding attribute. The protein concentration was measured by the BCA protein assay kit (Thermo Fisher Scientific, Waltham, MA, USA) as described in the instruction manual, before processing of immunogen formulation and immunization in the further step. 

### 2.4. Vaccine Immunizations and E. canis Challenge

BALB/c mice were used in this study for animal immunization and challenge infection. A total of 40 BALB/c mice, 6 to 8 weeks old were categorized into eight groups based on the immunogen formulations (*n* = 5 in each group). Groups 1 to 6, mice were immunized with rGP19 (0 µg, 50 µg and 100 µg, respectively) in Montanide adjuvant (Montanide™ ISA, SEPPIC, Paris, France), while groups 7 to 8 are non-immunized groups (control groups). Mice were intraperitoneally (i.p.) immunized, followed by a booster at 2 weeks from primary immunization. Except for groups 7 and 8, mice were challenged with *E. canis* (1 × 10^4^ *E. canis*) that were grown in DH82 cells. The challenge was performed after the booster immunization for 4 weeks by i.p. route and then observed daily. Five mice in each group were sacrificed on days 7 and 14 post-challenge (Table 1). Whole blood samples, sera, liver, and spleen were collected for further procedure. All procedures using mice including challenge exposure, and blood and organ collection were performed in the laminar flow biological cabinet of Biosafety level 2.

### 2.5. Measurement of rGP19 Antibody Elicits in Mice

Sera of all mice groups were collected pre- and post-*E. canis* challenging to determine antibody titers. An in-house indirect ELISA was performed, a concentration of 10 µg/ml rGP19 coated the immunoplates. The plates were incubated with the individual mice sera with the dilution of 1:1000. HRP conjugated rabbit anti-Mouse IgG (Invitrogen^TM^, Thermo Fisher Scientific, Waltham, MA, USA) was used as the secondary antibody at a dilution of 1:5000. Uncoated wells were used as blanks. The data were expressed as means with standard errors measured at the optical density (OD) of 450 nm.

### 2.6. Assessment of Ehrlichial Load in Organs by Real-Time PCR

Blood, liver, and spleen samples of the mice were obtained to extract the genomic DNA following the manufacturer’s instructions. Quantitative estimation of the ehrlichial load was performed using the quantitative PCR (qPCR) *E. canis*-16S rRNA, as performed in a previous study [14]. A total of 100 ng of DNA was used for each reaction. The qPCR reaction was performed using a SensiFAST SYBR Lo-ROX Kit (Bioline, London, UK) according to the manufacturer, and the reaction was the procedure under CFX96 Touch™ Real-Time PCR (Bio-Rad Laboratories, Hercules, CA, USA). A negative result for ehrlichial DNA was identified if the critical threshold values (Ct) of the qPCR reaction exceed 40 cycles. The expression analysis of the ehrlichial load was normalized relative to the total DNA.

### 2.7. Relevance of rGP19 on Cytokine mRNA Expression by qPCR

For investigation of the effects of rGP19 recombinant protein vaccine on cytokine expression, cytokine genes related to monocytes, macrophages, T cells, and B cells were quantitatively analyzed with qPCR as well. RNA in whole blood samples of each mouse was isolated and then used in cDNA synthesis for quantitative analysis of the mRNA transcriptions of gene coding that involved cytokine, including interferon-gamma (*IFNG*), interleukin 2 (*IL-1*), interleukin 4 (*IL-4*), interleukin 6 (*IL-6*), and tumor necrosis factor-alpha (TNF), with glyceraldehyde 3-phosphate dehydrogenase (*GAPDH*) as the housekeeping gene. The expression levels (fold-difference) were reported using the 2^−ΔΔCt^ method [15]. Specific information regarding the oligonucleotide sequences of primers used in this part of the study is presented in Table 2.

### 2.8. Preparation of Mice Splenocyte and Flow Cytometry Analysis

The frequencies of *E. canis* GP19-specific IFN-γ-producing T cells in mice spleens were investigated using flow cytometry modified from previous studies [16,17]. A number of 2 × 10^6^ cells of the splenocytes from individual mouse were isolated and then cultured in a 12-well culture plate in cell culture media, containing RPMI 1640 media (Gibco^TM^, Thermo Fisher Scientific, Waltham, MA, USA) supplemented with 10% inactivated fetal bovine serum (FBS; Gibco^TM^, Thermo Fisher Scientific, Waltham, MA, USA) and 1% antibiotic-antimycotic (Gibco^TM^, Thermo Fisher Scientific, Waltham, MA, USA) in the presence of three stimuli, including *E. canis* GP19 antigen (2 µg/mL), cultured media (negative control), or 1x cell stimulation cocktail (eBioscience™, San Diego, CA, USA) containing phorbol 12-myristate 13-acetate (PMA) and ionomycin (positive control). After 18 h of in vitro stimulation, the cells were harvested and then followed by 4 h of incubation with brefeldin A (BD GolgiPlug; BD Biosciences, San Diego, CA, USA). Fc receptors were blocked by adding 20 µL per test of anti-Fc II/III receptor monoclonal antibodies (MAbs) (BD Biosciences) to the sample and then incubate on ice for 20 min. Without washing, the splenocytes then were labeled with fluorescent-conjugated antibodies (eBioscience™, San Diego, CA, USA) for mice, including CD3 (APC; clone 17A2) for 0.5 μg/test, CD4 (FITC; clone RM4–5) for 0.25 μg/test, and CD8 (PerCP Cyn5.5; clone 53–6.7) for 0.25 μg/test of surface molecules. The cells were incubated on ice and protected from light for 60 min, and then washed two times with a flow cytometry staining buffer (cold PBS + 1% FBS) and centrifuged. The cells were resuspended in serum/protein-free PBS and labeled with a viability dye (LIVE/DEAD™ Fixable Dead Cell Stain Kits; Thermo Fisher Scientific, Waltham, MA, USA) on ice and protected from light for 30 min.

For intracellular IFN-γ straining, the splenocytes were fixed and stained with IFN-γ MAbs (PE; clone XMG1.2) for 0.25 μg/test. The sample acquisition (50,000 events) was performed on IFN-γ-containing CD4+ and CD8+ cells using a CyAnTM ADP Flow Cytometer (Beckman Coulter, Brea, CA, USA). The frequencies of antigen-specific IFN-γ-producing T cells in the spleens of the mice were investigated after the background staining of unstimulated cells was subtracted.

### 2.9. Data Analysis

The data in this study were analyzed for outliers and the normality test was performed prior to performing statistical analysis. The statistical analysis was performed using one-way ANOVA or the multiple t-test using GraphPad Prism 8.2.0 (GraphPad Software, Inc., San Diego, CA, USA). Multiple comparisons were analyzed using the Holm-Sidak method. Results of statistical analyses were considered significant in all experiments when *p* < 0.05. Results were reported as the mean and standard error (SE) or median value.

## 3. Results

### 3.1. Recombinant GP19 Protein (rGP19) Production

The purified rGP19 were examined and analyzed by Western blot. The rGP19 displayed the characteristic of reactive protein bands at a target molecular mass approximately 20–25 kDa (Figure 1; the original figure was provided as Appendix A). The result provided evidence of successful expression of rGP19 and the specificity of the HRP-conjugated antibody to the synthetic rGP19.

### 3.2. rGP19 Vaccine Prototypes Induce a Strong Antibody Response in Mice

Analysis of specific anti-rGP19 responses by ELISA on days 0, 7, and 14 after *E. canis* challenging demonstrated that the mean antibody levels in immunized mice (50 µg and 100 µg of rGP19) were statistically significantly higher (*p* < 0.01) than those immunized with only adjuvant and naïve mice, pre- and post-challenge with *E. canis* (Figure 2).

With the results, antibody response induced by the rGP19 vaccine prototypes was associated with the development of antigen-specific antibody responses.

### 3.3. rGP19 Vaccine Prototypes Provide the Protection against the Pathogen

There was no evidence of bacterial loads in the naïve mice groups (groups 7 and 8). The protection against *E. canis* was evaluated with *E. canis* challenging and determined the bacterial copy number by real-time PCR at different days post-challenge. The results showed lower (*p* < 0.01) *E. canis* load in blood, liver, and spleen on days 7 and 14 post-challenge in the immunized mice (both 50 µg and 100 µg) compared with the adjuvant-immunized mice (Figure 3).

### 3.4. Relevance of rGP19 on Cytokine-Relative Gene Expression by qPCR

Cytokine gene expression levels in naive mice and mice challenged with *E. canis* were investigated, including *IFNG*, *IL1a*, *IL4*, *IL-6,* and *TNF* genes. On day 7 post-infection period, cytokine gene expression levels showed overall no different expression among the mice group, except for *IFNG*. The immunized mice with 100 µg of rGP19 and adjuvant showed a higher expression level of *IFNG* than other groups (*p* < 0.05). 

On day 14 post-infection period, the immunized mice with rGP19 (both 50 µg and 100 µg combined with Montanide adjuvant) and later infected with *E. canis* showed significantly up-regulated expression of *IFNG* (*p* < 0.01), *IL1a* (*p* < 0.01), and *IL4* (*p* < 0.05) compared with naive mice on day 14 post-infection, as shown in Figure 4.

The results revealed that the immunized mice with Montanide adjuvant and later infected with *E. canis* had significantly higher *TNF* gene expression (*p* < 0.01) than other mice groups 14 days after infection. In contrast, *IL6* gene expression of adjuvant immunized, and later infected mice showed down-regulation compared with other mice groups.

### 3.5. rGP19 Manipulated Specific Memory CD4+ Th1 Responses during E. canis Infection

The immunized and non-immunized mice displayed no difference in the frequencies of IFN-γ-producing CD8+ T cells in splenocytes after being stimulated with the *E. canis* rGP19 antigen. In the part of CD4+ T cells, there is no difference among mice splenocytes that were stimulated with PMA, cell media, or *E. canis* rGP19 antigen on day 7 post-infection (Figure 5A). 

However, there are significantly different (*p* = 0.011) splenocytes on day 14 post-infection when stimulated *E. canis* antigen on day 14 post-infection. Compared with uninfected na-ive mice, the immunized mice (with 50 µg and 100 µg of rGP19) had significantly higher frequencies of IFN-γ-producing CD4+ T cells in splenocytes (Figure 5B and Figure 6).

## 4. Discussion

GP19 of *E. canis* was found to have orthologs with variable-length PCR target proteins (VLPT) of *E. chaffeensis*, but it lacked the tandem repeats (TRs) that were present in *E. chaffeensis* VLPT [11]. Moreover, the GP19 protein exhibited O-linked glycosylation sites that were stated in the STE-rich patch. The amino-terminus of the STE-rich patch consisted of an epitope recognized by the host [11]. Although this protein seems to play an important role in host-microbe interaction, the information of the application using GP19 as a vaccine candidate against *E. canis* infection is not available. 

In this present study, the ability of recombinant GP19 protein vaccine in protecting mice against *E. canis* infection was analyzed. The rGP19 at the molecular mass of approximately 20–25 kDa was used as a prototype vaccine against *E. canis* infection in mice. The protection against *E. canis* infection after challenge exposure for 7 days and 14 days was explored and revealed that the mean antibody levels in immunized mice (50 µg and 100 µg of rGP19) were statistically significantly higher than immunized with only adjuvant and naive mice, pre- and post-challenge with *E. canis*. Interestingly, the elevated antibodies in rGP19-immunized mice were along with the lower ehrlichial load in blood, liver, and spleen on days 7 and 14 post-challenge compared with the adjuvant-immunized mice detected by qPCR. The results of the protection role by reduction in *E. canis* in the rGP19-immunized mice in this study showed similarity to the previous studies of the protection role of P28 against *Ehrlichia* in mice, *E. cheffeensis,* and *E. muris* [16,18,19,20]. A previous study demonstrated the spontaneous clearance of *E. cheffeensis* infection in the BALB/c mice that were immunized with the rP28 [18]. Moreover, the results in this study supported the possibility of humoral immunity role in the part of the protection in mice challenged with *E. canis* in the rGP19-immunized mice. 

The activation of phagocytes and cell-mediated immune response plays major roles against *Ehrlichia* with various soluble cell products [21,22]. There was the possibility of rGP19 provoked CD4+ memory T cell responses due to the detection of the antigen-specific IFN-γ-producing memory CD4+ T cells in the rGP19-immunized mice later infected with *E. canis* on day 14 post-infection period. The results at the cellular level in this study provided evidence of rGP19 that can eliminate *E. canis* by manipulating Th1 and B cell roles, similar to the previous studies that determined the effect of structural-based peptide and recombinant P28 against *E. muris* [16,17]. 

Cytokines are cell-signaling molecules that are synthesized by different immune cells. Specific receptors on the target cells are bonded with the cytokines, and then trigger signal-transduction pathways that are involved in innate and adaptive immunity to defend against pathogens or disease conditions [23,24]. Accordingly, cytokines are imperative for the phenomenon of immunity, including their role in developing and regulating [25]. The results in the part of in vivo of this study provide a perspective on *IFNG* genes that involved macrophage-Th1 cells, according to in vitro study. The *IFNG* expression of immunized mice with 100 µg of rGP19 displayed significantly up-regulated levels since day 7 post-challenge of *E. canis*. The result confirmed the possibility that *IFNG* expression in the rGP19-immunized mice infected with *E. canis* can be related to the elimination of the microorganism, *E. canis*. 

IL-4 is one of the cytokines involved with the Th2 response. The inactivated vaccine of *Ehrlichia ruminantium* provided the elicit of CD4+ and CD8+ to produce IFN-γ; however, there was an absence of IL-4 in a previous study [26]. The results from the previous study indicated the type 1 response that was influenced by the inactivated vaccine [26]. Contrastingly, the *IL4* gene expression of the 100 µg of rGP19-immunized mice later infected with *E. canis* in vivo in this study displayed a positive relationship with *IFN* on day 14 post-infection period. The animal model showed the possibility that rGP19 might stimulate both Th1 and Th2 responses. Thus, the investigation of the cell-mediated immune response including type 1 and 2 cytokines in immunization with the GP19 peptide and protein in animal models is further needed for the full understanding of the protective ability.

TNF-α is a pro-inflammatory cytokine produced predominantly by macrophages [27,28]. The *TNF* gene in mouse blood samples showed up-regulated expression in the *E. canis*-infected mice with the absence of rGP19 immunization on day 14 of the post-infection period. In addition, *IL6* gene encoding of IL-6 is one of the pro-inflammatory and anti-inflammatory myokines mediated through its inhibitory effects on TNF-alpha activation, which showed a negative relationship with *TNF* expression. Importantly, IL-1 which is expressed in various immune cells, including macrophages, monocytes, dendritic cells, B cells, NK cells, and epithelial cells, promotes adaptive immunity [29,30]. In this study, rGP19 non-immunized and immunized mice with/without *E. canis* challenge were investigated in the *IL1a* encoding IL-1α. The rGP19-immunized mice (50 µg and 100 µg) later infected with *E. canis* showed up-regulation of *IL1a* compared with naive mice on day 14 post-infection period, according to the T cell responses examined by flow cytometry. However, the relationship between cytokine genes and adaptive immune response needs further study for the rational development of vaccines.

Our analyses of the cytokine networks suggest that there are supplementary cytokines involved in the *E. canis* infection responses with the cooperation of macrophages and lymphocytes (T and B cells). The possible linkages of cytokine expressions resulting from macrophages and cell-mediated immune response against intracellular microorganisms including *E. canis* are shown in Figure 7.

## 5. Conclusions

The limitation of vaccine development against *E. canis* infection is that the knowledge of the protective immunity of the host is still unknown. In this study, recombinant protein GP19 was produced as a vaccine prototype, rGP19, for inducing protective immune responses in a mouse model against *E. canis*. Antibody responses against *E. canis* were evaluated and revealed that the antibody levels in rGP19-immunized mice were significantly higher than adjuvant-immunized and naive mice, pre- and post-*E. canis* challenging. DNA from blood, liver, and spleen were extracted to determine ehrlichial loads. The rGP19-immunized mice showed significantly lower ehrlichial loads in blood, liver, and spleen DNA compared with adjuvant-immunized mice. This study also provided the detection of IFN-γ-producing memory CD4+ T cells in the rGP19-immunized mice later infected with *E. canis* on day 14 post-infection period using flow cytometry. Additionally, Cytokine expression was investigated and revealed up-regulation of IFNG and IL1 mRNA expression in rGP19-immunized mice. The present studies provided evidence of rGP19 that can eliminate *E. canis* by manipulating both humoral and phagocyte cell-mediated immune responses in a laboratory animal model.

## Figures and Tables

**Figure 1 vetsci-09-00386-f001:**
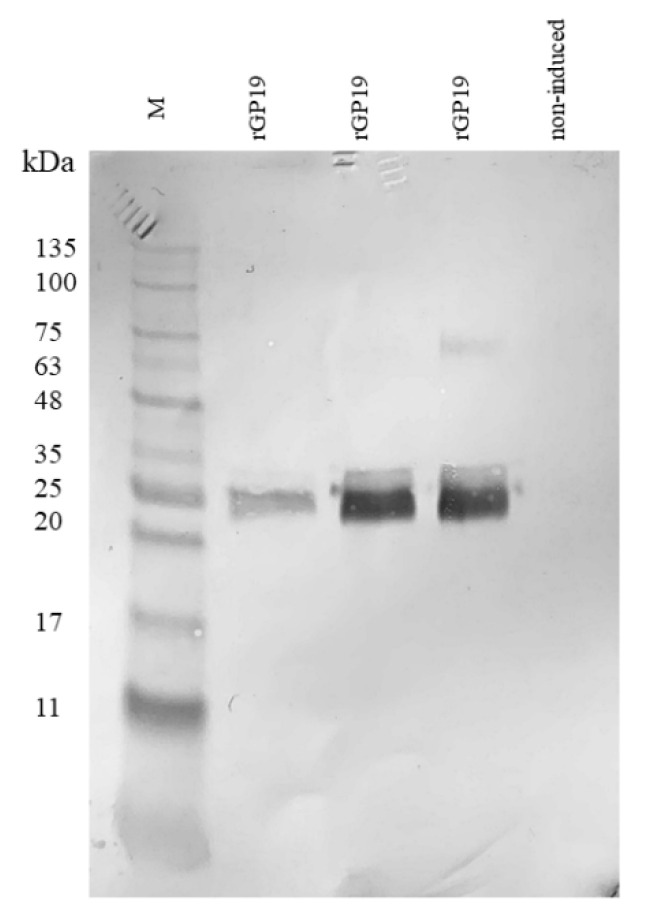
Recombinant protein of *Ehrlichia canis* (rGP19) expression was affirmed with the HRP conjugated IgG antibody against rGP19 in Western blot analysis. Lane M: protein marker; rGP19: recombinant protein GP19 induced with IPTG; non-induced: non-induced rGP19 with IPTG.

**Figure 2 vetsci-09-00386-f002:**
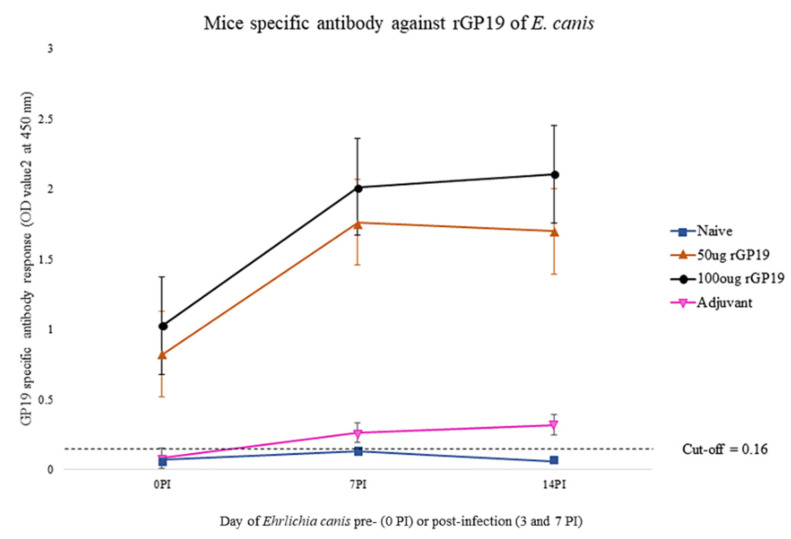
The association between antibody responses induced by the rGP19 vaccine and the development of antigen-specific antibody responses, pre- and post-*E. canis* infection in mice. rGP19 vaccine prototypes provide the protection against the pathogen measured by an ELISA using a purified rGP19 as the antigen on day 0 (pre-*Ehrlichia canis* challenging), and days 7 and 14 (post-*E. canis* challenging).

**Figure 3 vetsci-09-00386-f003:**
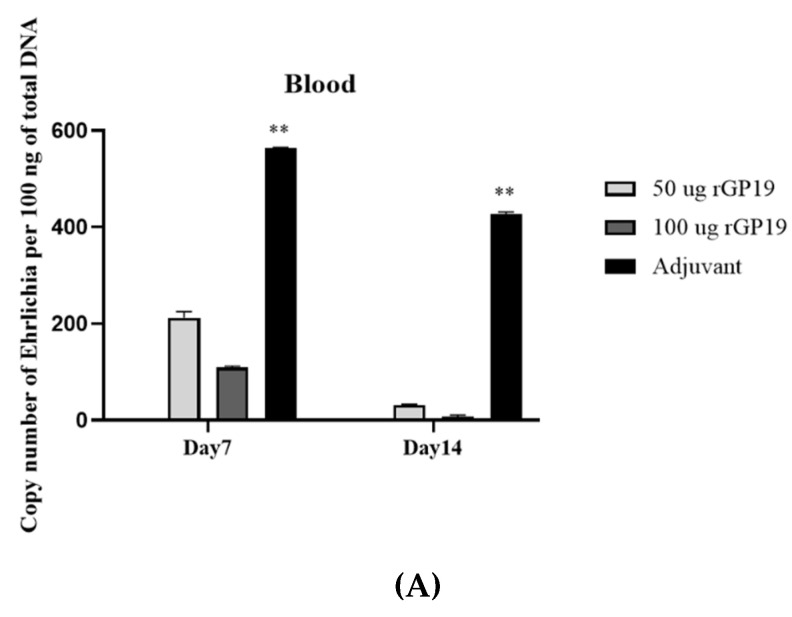
Immunization with rGP19 provided protection against *Ehrlichia canis* infection in mice. The protection against *E. canis* infection in the immunized mice with rGP19 was determined by the ehrlichial load in the blood (**A**), liver (**B**), and spleen (**C**) samples measured by quantitative PCR (qPCR) and showed lower ehrlichial loads on days 7 and 14 post-challenge (** *p* < 0.01) compared with mice immunized with adjuvant.

**Figure 4 vetsci-09-00386-f004:**
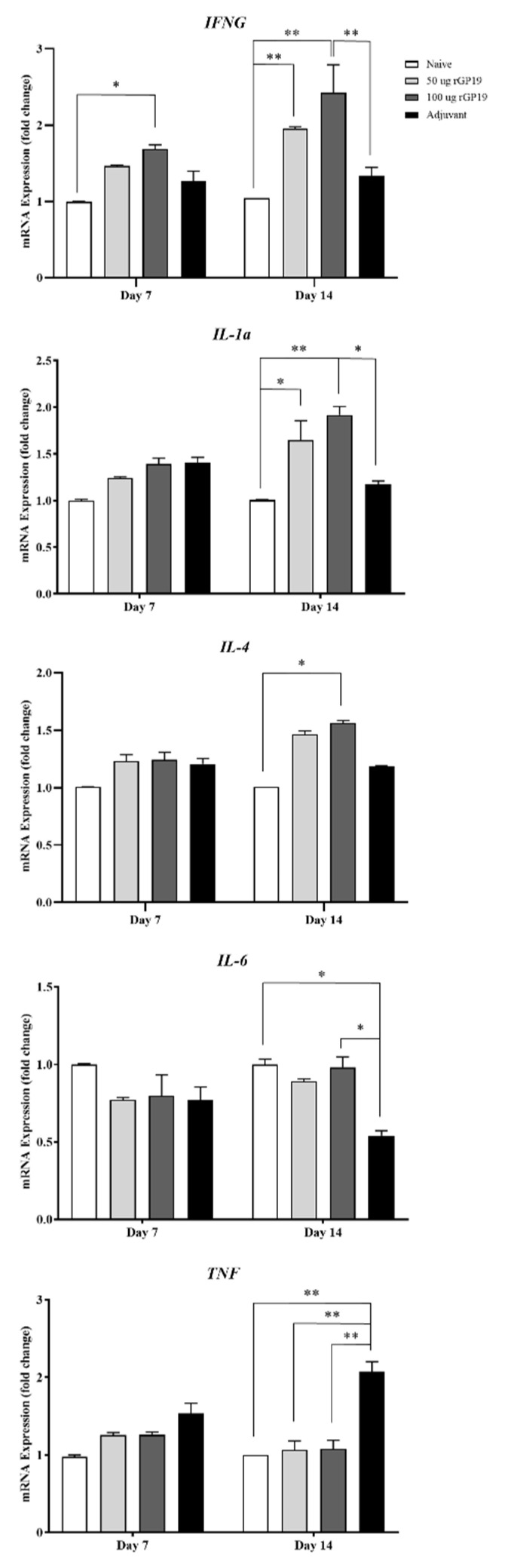
Analyses of cytokine mRNA expression in mice using qPCR. Bar graphs represent relative *IFNG*, *IL-1a*, *IL-4*, *IL-6*, and *TNF* mRNA expression levels after normalization to the *GAPDH* gene in naive and immunized mice with later *Ehrlichia canis* challenge group (50 µg rGP19, 100 µg rGP19, and adjuvant). Data are represented as mean ± SE (*n* = 5 each treatment), one-way ANOVA, * *p* < 0.05, ** *p* < 0.01.

**Figure 5 vetsci-09-00386-f005:**
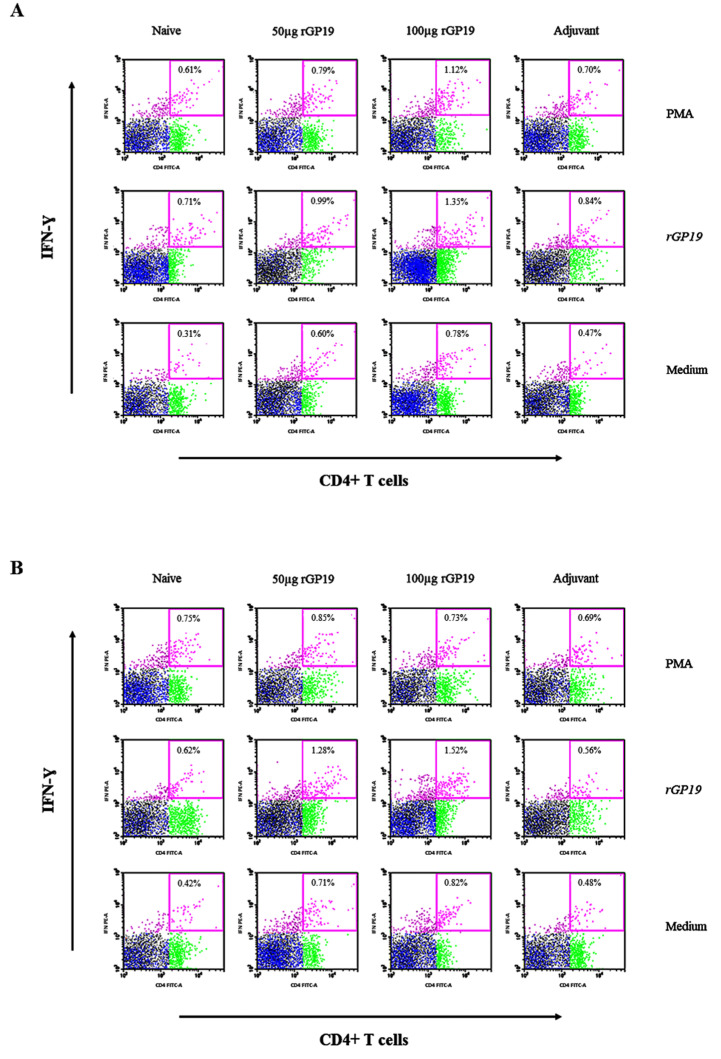
*Ehrlichia canis* GP19-specific IFN-γ-producing CD4+ T cells develop during *E. canis* infection in mouse splenocyte determined by flow cytometry. Naive and immunized mice infected with *E. canis* over a period of 7 days (**A**) and 14 days (**B**) post-infection. The data plot represents the populations of lymphocytes (black dot), CD3+ T lymphocytes (blue dot), CD4+ T lymphocytes (green dot), and IFN-γ molecules in CD4+ T lymphocytes (pink dot).

**Figure 6 vetsci-09-00386-f006:**
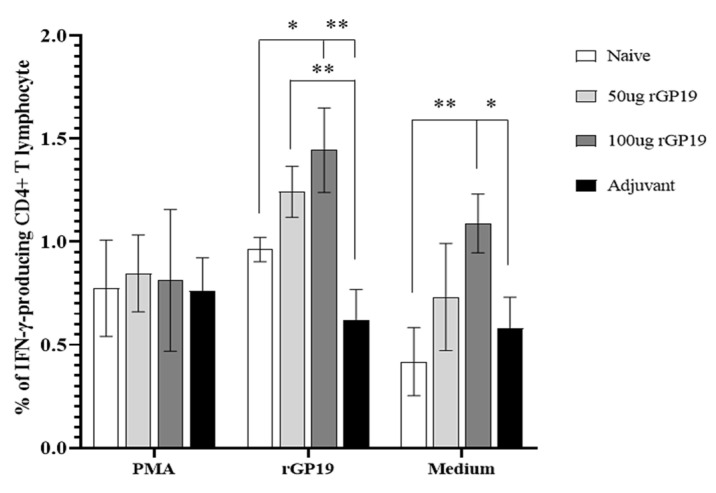
Mean of the frequencies of IFN-γ-producing CD4+ T lymphocytes in the splenocytes of mice on days 14 post-*Ehrlichia canis* infection when stimulated with different treatments. Data are represented as mean ± SE, one-way ANOVA, * *p* < 0.05, ** *p* < 0.01.

**Figure 7 vetsci-09-00386-f007:**
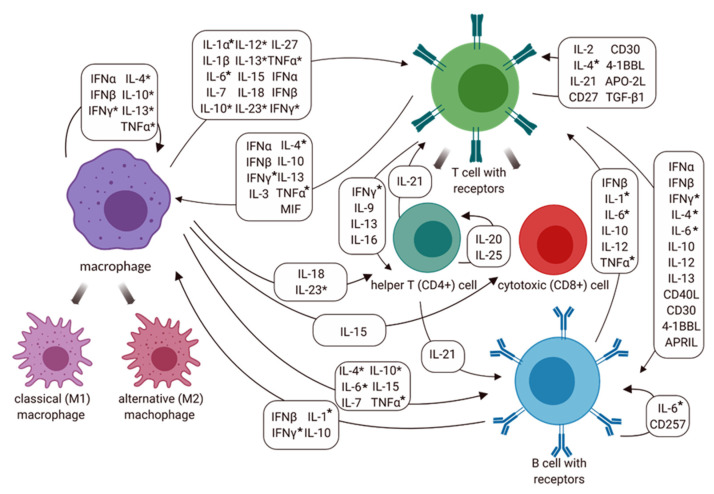
Cytokine network involved macrophage, T cell (CD4+ and CD8+) and B cell activation in the immune response against *Ehrlichia canis*. Abbreviations: 4-1BBL: 4-1BB ligand; APRIL: a proliferation-inducing ligand; CD: cluster of differentiation; IFN: interferon; IL-: interleukin; MIF: macrophage migration inhibitory factor; TNF: tumor necrosis factor; TGF: transforming growth factor. Additionally, * represents the cytokine expression at the molecular level in this study and previous study [13].

**Table 1 vetsci-09-00386-t001:** Immunization and *Ehrlichia canis* challenge experiments in mice.

Group	Number of Mice/Group	Immunogen Formulation	Challenge Exposure	Day of Euthanasia Post-Infection (PI)
1	5	Montanide (50 µL) *	✓	7 PI
2	5	Montanide (50 µL) *	✓	14 PI
3	5	50 µg rGP19 protein (200 µL) with Montanide (50 µL) *	✓	7 PI
4	5	50 µg rGP19 protein (200 µL) with Montanide (50 µL) *	✓	14 PI
5	5	100 µg rGP19 protein (200 µL) with Montanide (50 µL) *	✓	7 PI
6	5	100 µg rGP19 protein (200 µL) with Montanide (50 µL) *	✓	14 PI
7	5	No immunization (control)	🗶	7 PI
8	5	No immunization (control)	🗶	14 PI

* Intraperitoneal administration. ✓ The challenge exposure with *E. canis* was performed. 🗶 The challenge exposure with *E. canis* was not performed.

**Table 2 vetsci-09-00386-t002:** PCR and oligonucleotide sequences of primers targeting cytokine genes used in this experiment.

Primer Name	Oligonucleotide Sequence (5′–3′)	Target Gene	Product Size
mIFNG-F mIFNG-R	5′-TAT TGT CGC TTC TGG CTC CT-3′ 5′-AGA CTT ACG GCT GGC TTT GA-3′	*IFNG*	227
mIL1a-F mIL1a-R	5′-TCA AAG CCC AAA GGA AGC TA-3′ 5′-AGC TGA CTG CTC TGG GGA TA-3′	*IL-1a*	177
mIL4-F mIL4-R	5′-CTG GGC TTT CCT AGC TGA TG-3′ 5′-CTC TGT GGG GCA ATA CCT GT-3′	*IL-4*	214
mIL6-F mIL6-R	5′-CAG AGG ATA CCA CTC CCA ACA-3′ 5′-TCC AGT TTG GTA GCA TCC ATC-3′	*IL-6*	203
mTNF-F mTNF-R	5′-CAT GCG TCC AGC TGA CTA AA-3′ 5′-TCC CCT TCA TCT TCC TCC TT-3′	*TNF*	182
mGAPDH-F mGAPDH-R	5′-ACC CAG AAG ACT GTG GAT GG-3′ 5′-CAC ATT GGG GGT AGG AAC AC-3′	*GAPDH*	171

## Data Availability

All data in this study are included in the manuscript.

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
