# Peer review of "Recombinant Ehrlichia canis GP19 Protein as a Promising Vaccine Prototype Providing a Protective Immune Response in a Mouse Model"

_vetsci, 2022, doi:10.3390/vetsci9080386_

Round 1

Reviewer 1 Report

The manuscript is very interesting and well written. My only question is why was it all done in mice and not in dogs? and why was E. canis used in mice? Explain it in the text.

Author Response

Authors’ Responses to Reviewer(s)' Comments:

Manuscript ID: vetsci-1805658

Manuscript title: Recombinant Ehrlichia canis GP19 protein as a promising vaccine prototype providing a protective immune response in a mouse model

Journal: Veterinary Sciences

Section: Veterinary Microbiology, Parasitology and Immunology

Response to reviewers’ comments:

The authors would like to thank all reviewers for their kind consideration and review. The authors have carefully reviewed all comments and have revised the manuscript accordingly. Our responses to the comments are given below.

Reviewer comments:

Reviewer #1: The manuscript is very interesting and well written. My only question is why was it all done in mice and not in dogs? and why was E. canis used in mice? Explain it in the text.

Response: The authors would like to thank the reviewer for consideration and kind review. The authors would like to inform you that we have a plan to evaluate the efficacy of recombinant vaccine candidates in the definitive host, dog, after the previous in vitro experiment (Nambooppha et al., 2021). However, we have faced the issue of using dogs as laboratory animals in the animal ethical system. The Laboratory Animal Use ethics required for alternative animal testing in order to use the results of this experiment in an application for further use of dogs as laboratory animals.

            In addition, there were shreds of evidence of E. canis infection has been reported in wild mice (Kawahara et al., 1993), and successful propagation of E. canis on BALB/C mouse macrophages (Keysary et al., 2001). Therefore, we determined the efficacy of the vaccine prototype against E. canis infection in BALB/C mice in this study, in place of dogs that show clinical signs. The present study provided evidence of rGP19 that could eliminate E. canis in mouse model and provide the possibility of the development of vaccines to provide protection against E. canis infection in the definitive host for the further study.

Supplementary references:

Nambooppha, B.; Rittipornlertrak, A.; Muenthaisong, A.; Koonyosying, P.; Tangtrongsup, S.; Tiwananthagorn, S.; Chung, Y.T.; Sthitmatee, N. Effect of GP19 peptide hyperimmune antiserum on activated macrophage during Ehrlichia canis infection in canine macrophage-like cells. Animals. 2021, 11, 2310.

Kawahara, M.; Suto, C.; Rikihisa, Y.; Yamamoto, S.; Tsuboi, Y. Characterization of ehrlichial organisms isolated from a wild mouse. J. Clin. Microbiol. 1993, 31, 89-96.

Keysary, A.; Waner, T.; Strenger, C.; Harru,s S. Cultivation of Ehrlichia canis in a continuous BALB/C mouse macrophage cell culture line. J. Vet. Diagn. Invest. 2001, 13, 521-523.

Other amendments:  The suggested modification has been applied in the revised manuscript (indicated by blue highlighting).

Reviewer 2 Report

This is an interesting study using a recombinant protein vaccine against Ehrlichia canis to vaccinate mice. It is generally well done and well reported but could be improved a little, as under:

1. the written English is poor in parts (eg lines 50/51, 116, 250) and should be reviewed by a native English writer.

2. line 109. What is the BCA protein assay ? (reference needed ?)

3. line 112. State how many mice were in each of the  experimental groups. Control groups stated as n=5.

4.  line 113. What is "Montanide" adjuvant ? Reference needed ?

5. In Figure 3 I can't see any bars corresponding to "naive" mice. Remove this from the 4 coding figures.

6. Figure 6 is confusing. There is too much data bunched up. I can't see the differences properly.

7. Are you panning to test your vaccine in dogs ? Just because it works in experimental mice does not mean it will necessarily work in dogs, the natural host animal for E.canis.

Author Response

Authors’ Responses to Reviewer(s)' Comments:

Manuscript ID: vetsci-1805658

Manuscript title: Recombinant Ehrlichia canis GP19 protein as a promising vaccine prototype providing a protective immune response in a mouse model

Journal: Veterinary Sciences

Section: Veterinary Microbiology, Parasitology and Immunology

Response to reviewers’ comments:

The authors would like to thank all reviewers for their kind consideration and review. The authors have carefully reviewed all comments and have revised the manuscript accordingly. Our responses to the comments are given below.

Reviewer #2: This is an interesting study using a recombinant protein vaccine against Ehrlichia canis to vaccinate mice. It is generally well done and well reported but could be improved a little, as under:

Comment 2.1: The written English is poor in parts (eg lines 50/51, 116, 250) and should be reviewed by a native English writer.

Response: The authors would like to thank the reviewer for pointing this out. The manuscript was appropriately revised, lines 62-64, 130-134, 264-272 (indicated by green highlighting).

Comment 2.2: line 109. What is the BCA protein assay? (reference needed?)

Comment 2.3:  line 112. State how many mice were in each of the experimental groups. Control groups stated as n=5.

Response: The suggested modification has been applied in the Materials and Methods section of the revised manuscript (lines 125-126) indicated by green highlighting.

Comment 2.4:  line 113. What is "Montanide" adjuvant? Reference needed?

Response: Montanide is an oil-based adjuvant composed of natural metabolizable oil and a very refined emulsifier. Compared to traditional oil emulsions, Montanide emulsions are stable and easy to inject with high immunopotentiation capacity and show lesser side effects in animal experiments (Aucouturier et al., 2001, Khabazzadeh et al., 2016). Consequently, we used the Montanide adjuvant to produce the immunogen formulations in this study.

As has been concerned by the reviewer, the reference of Montanide adjuvant has been added in the revised manuscript at lines 127-128 (indicated by green highlighting).

Supplementary references:

Aucouturier, J.; Dupuis, L.; Ganne, V. Adjuvants designed for veterinary and human vaccines. Vaccine. 2001, 19, 2666-2672.

Khabazzadeh, T.N.; Mahdavi, M.; Maleki, F.; Zarrati, S.; Tabatabaie, F. The role of Montanide ISA 70 as an adjuvant in immune responses against Leishmania major induced by thiol-specific antioxidant-based protein vaccine. J. Parasit. Dis. 2016, 40, 760-767.

Comment 2.5: In Figure 3 I can't see any bars corresponding to "naive" mice. Remove this from the 4 coding figures.

Response: The authors apologize that the figure was not clear enough and thank the reviewer for pointing out this point. Firstly, we would like to inform you that the DNA was extracted from all groups of mice, including naïve groups (as negative control groups that cannot be amplified the Ehrlichial gene) for quantitative estimation of the E. canis load by qPCR. Since there was no evidence of Ehrlichial DNA in the naïve groups, no bars were displayed in Figure 3. The “Results” in 3.3. rGP19 vaccine prototypes provide the protection against the pathogen (line 233) and also Figure 3 were revised for clearer understanding in the revised manuscript (indicated by green highlighting).

Comment 2.6: Figure 6 is confusing. There is too much data bunched up. I can't see the differences properly.

Response: The authors would like to thank the reviewer for pointing out this point. Figure 6 and the caption (lines 280-282) were revised to show a clearer significant difference (indicated by green highlighting).

Comment 2.7: Are you planning to test your vaccine in dogs? Just because it works in experimental mice does not mean it will necessarily work in dogs, the natural host animal for E.canis.

Response: Since we have faced the issue of using dogs as laboratory animals in the animal ethical system that required alternative animal testing in order to use the results of this experiment in an application for further use of dogs as laboratory animals. There were shreds of evidence of E. canis infection has been reported in wild mice and successful propagation of E. canis on BALB/C mouse macrophages, therefore, we determined the efficacy of the vaccine prototype against E. canis infection in BALB/C mice in this study, in place of dogs that show clinical signs.

We have a plan to evaluate the efficacy of recombinant vaccine candidates in the dog, definitively. The present study provided evidence of rGP19 that could eliminate E. canis (in mouse model) and provide the possibility of the development of vaccines to provide protection against E. canis infection in the definitive host for further study.
